# Quantum Dots and Their Interaction with Biological Systems

**DOI:** 10.3390/ijms231810763

**Published:** 2022-09-15

**Authors:** Nhi Le, Min Zhang, Kyoungtae Kim

**Affiliations:** 1Department of Biology, Missouri State University, 901 S National, Springfield, MO 65897, USA; 2Department of Toxicology and Cancer Biology, University of Kentucky, Lexington, KY 40506, USA

**Keywords:** quantum dots, mammalian, fungal, plants, interaction, proteins

## Abstract

Quantum dots are nanocrystals with bright and tunable fluorescence. Due to their unique property, quantum dots are sought after for their potential in several applications in biomedical sciences as well as industrial use. However, concerns regarding QDs’ toxicity toward the environment and other biological systems have been rising rapidly in the past decade. In this mini-review, we summarize the most up-to-date details regarding quantum dots’ impacts, as well as QDs’ interaction with mammalian organisms, fungal organisms, and plants at the cellular, tissue, and organismal level. We also provide details about QDs’ cellular uptake and trafficking, and QDs’ general interactions with biological structures. In this mini-review, we aim to provide a better understanding of our current standing in the research of quantum dots, point out some knowledge gaps in the field, and provide hints for potential future research.

## 1. Introduction

Due to their potential in applied science, quantum dots and their toxicity have been intensively studied and reviewed in the past few decades [1,2,3,4,5,6,7]. Quantum dots (QDs) are nanosized (2–10 nm) semiconductor crystals with distinguished chemical and physical properties, enabling them to emit a wide range of bright, photobleaching-resistant light [8]. QDs’ fluorescence is size tunable, which allows for simple adjustment of QDs’ size and composition to achieve desired color [5,9,10,11,12,13,14,15,16,17,18,19,20]. It was found that increased diameter of QDs caused a redshift in fluorescence [17]. Additionally, quantum dots with greater height in dimension display longer photoluminescence lifetime and increased emission wavelength [15]. Thus, larger quantum dots emit more stable fluorescence in higher emission ranges. In addition to QDs’ size, the shape of QDs also significantly influence QDs’ stability and optical property [19,20,21,22,23]. Some of the most used quantum dot shapes include spherical QDs, cylindrical QDs, pyramidal QDs, conical QDs, tetrahedral QDs, and lens-shaped QDs [19,20,21,24]. It was found that tetrahedral QDs have sharper edge absorption and are more confined than spherical QDs. On the other hand, spherical QDs at 3.1 nm showed to be most efficient in photon absorption and required less excitation energy compared to other types of QD shapes [19,21].

Quantum dots are typically sorted based on their core type, shape, structure, size, and ligands. Common quantum dot cores include cadmium, indium, and carbon encapsulated by chalcogenides such as selenides, tellurides, and sulfide [25,26,27,28,29]. However, core-only quantum dots have been shown to be unstable due to the deterioration of materials [30,31]. For core-type QDs such as cadmium selenide quantum dots (CdSe QDs), an oxidizing environment causes oxidation of the selenide (Se) layer on the QDs’ surface, thus weakening the overall structure of QDs and leading to leakage of cadmium ions. When adding an additional shell layer such as zinc sulfide (ZnS), the oxidation of Se is reduced, thus decreasing the amount of Cd ion leakage overall [30]. Compared to core-only QDs, quantum dots with a core–shell structure are considered superior when it comes to structural stability and photoluminescent quantum yield (PLQY) [31,32,33]. It is known that long-term exposure to environmental factors such as blue light and UV light quenches QDs’ fluorescence. The addition of a protective shell has shown to be effective in increasing QD resistance against photobleaching [34,35,36,37,38]. In addition to increased QDs’ stability, the presence of an exterior shell has been found to reduce the toxicity of quantum dots, which broadens their application range [39]. Therefore, a new type of quantum dot called core–shell quantum dots has been developed and is currently widely used. The shell of some quantum dots is decorated with ligands that provide further stability [40,41,42]. The addition of ligands allows QDs to have specific interactions with various environmental factors, which is useful in certain applications such as biosensing and particle detection [43,44,45,46]. Appropriate ligand choice could also aid the dispersion of QDs in an aqueous solution, thus minimizing the aggregation of QDs and resulting in accurate emission of size-dependent QDs [47].

Due to their unique characteristics, quantum dots have become a promising candidate for a range of important applications. Quantum dots are sought after for their potential in biomedical science, particularly for biosensing, drug delivery, cell tracking, disease detection, and potential antimicrobial/antibiotic remedies [48,49,50,51,52,53]. Furthermore, quantum dots are currently heavily utilized in several commercialized products, such as electronic devices, solar cells, LEDs, cosmetics, plastics, and other products essential to daily life [54,55,56,57]. As quantum dots gained attention in recent years, a few studies have shown the potential toxicity of quantum dots to mammalian cells, fungal cells, plants, and other organisms [25,58,59,60,61,62,63]. Although advances have been made, the limited knowledge of the effect of quantum dots on the environment and human health raises concerns about the use of quantum-dot-based products [64,65]. The findings regarding QD toxicity have previously been well reported in several review papers, however, none described QDs’ interactions with different biological living systems [2,3,4,5,6,7,66]. In this mini-review, we focus on the interactions between QDs and three types of biological systems: mammalian cells, fungal cells, and plant cells. We also aim to provide the most up-to-date advances in quantum dot research and propose a list of future research trends.

## 2. Core-Type Quantum Dots

As of today, a wide variety of quantum dots have been developed, each with slightly different properties and potential applications [13,32,58,67,68,69,70,71,72,73,74]. Some of the most common quantum dots that have captured current interest in the scientific community are cadmium QDs, indium QDs, graphene QDs, and carbon dots [26,56,63,72,75,76,77,78,79,80,81,82]. Each of these types of quantum dots possess a unique set of advantages and disadvantages, which make them suitable for slightly different applications (Figure 1). Among these, cadmium-based quantum dots are the most intensively researched.

Due to their high photoluminescence yield and environmental stability, cadmium-based quantum dots have been widely utilized in biomedical and photovoltaic technology [25,75], in which cadmium selenide quantum dots (CdSe QDs) are more frequently used compared to other cadmium-based QDs, such as cadmium sulfide and cadmium telluride, due to the wider visible fluorescence range [23]. For this reason, CdSe QDs are also more well investigated compared to the other cadmium-based QDs. Due to their superior quantum yield, previous work has suggested that cadmium QDs have vast potential in biomedical applications. Yet, there have been reservations regarding in vivo usage due to their high toxicity profile [18,55,56,82,83,84]. It has been assumed that the toxicity of cadmium quantum dots likely stems from the toxic cadmium ion content [23], and, therefore, research for a cadmium-free alternative has been ongoing in recent years [49,85,86].

Previously, a few studies have suggested that indium-based quantum dots, particularly indium phosphide zinc sulfide (InP/ZnS), could be an alternative to cadmium-based quantum dots [87,88]. It was found that with a similar amount of core leakage, InP/ZnS quantum dots seem to induce less cell damage than CdSe/ZnS, suggesting that InP (III) might be less toxic than cadmium ions [89]. Similar to that previously observed in cadmium QDs, the zinc shell has also been shown to play a crucial role in limiting InP-based QD toxicity [90]. Although considered inferior to cadmium-based quantum dots in fields such as bioimaging and LED performance due in part to lower quantum yield [91], the reduction in cytotoxicity makes indium-based quantum dots one of the most promising candidates for biomedical and industrial applications. Yet, Horstmann et al. showed that with the same amount of treatment concentration (100 µg/mL), InP/ZnS QDs inhibited yeast growth while CdSe/ZnS QDs prolonged the cell’s lag phase without affecting its final optical density [76]. Thus, the proposal of using InP-based quantum dots as an alternative to cadmium-based quantum dots needs additional assessments.

Recently, a group of heavy-metal-free quantum dots called carbon dots have gained attention due to their promising future in biological applications [20,80,92]. These carbon dots are comprised of two main categories: carbon quantum dots and graphene quantum dots. Carbon quantum dots are sphere-shaped and are composed of widely available and eco-friendly carbon atoms, which make them less toxic than traditional semiconductor quantum dots. Due to their environmental compatibility, carbon dots have been the most recently proposed alternative to traditional quantum dots [93,94,95,96]. In the last few years, the ability to synthesize carbon dots from biomass and waste products has also sparked researchers’ interest in using carbon dots in recycling and waste management [97,98,99]. However, studies have shown that although they have a low toxicity impact, carbon dots still induced shrinkage and the formation of holes on the surface of yeast cells [100,101]. Thus, it would be unwise to categorize carbon dots as safe before further investigation. Graphene quantum dots (GQDs), on the other hand, consist of a single layer of carbon atoms arranged in a two-dimensional honeycomb shape [66,77]. Most current studies on GQDs agreed that GQDs have a low toxicity effect, even in vivo testing [102,103,104]. Therefore, the appeal of using GQDs for biomedical science and disease treatment has been the focus of research for this type of QD [105,106,107,108,109]. In addition, carbon-based QDs are also desired for energy-related applications such as protogalactic energy conversion and super capacitator production [110,111]. Despite the potential in low toxicity use for biological processes, the research on GQDs is still in its infancy and needs further understanding before being commercially and industrially employed.

## 3. Quantum Dot Interaction with and Impact on Mammalian Cells

### 3.1. Interaction and Intracellular Trafficking

As the potential to utilize QDs in disease treatment and biological applications arises, the need to understand QDs’ interaction with all living systems, especially mammalian cells, has also become increasingly important. Thus, an intensive amount of effort has been made to map the behavior of QDs in mammalian cells (Table 1). Recent studies have suggested that interactions between different types of mammalian cells and QDs result in dissimilar entry pathways and intracellular trafficking of QDs. In general, QDs first interact with a selectively permeable cell membrane layer [112], which divides the intracellular space from its environment as well as controls the transit of materials into and out of the cell. The cell plasma membrane has a variety of uptake mechanisms [113,114,115], among which QDs were found to internalize into mammalian cells using multiple pathways, including clathrin-mediated endocytosis, caveolae-mediated endocytosis, and micropinocytosis pathways [116,117,118,119,120,121].

Lines of evidence have suggested that quantum dots aggregate on the surface of cells by binding with cell surface receptors, and then are internalized by receptor (or clathrin)-mediated endocytosis [122,123,124]. In general, QDs are mainly transported into mammalian cells by clathrin-mediated endocytosis [117,119,120,125]. However, some cells use a combination of pathways to take up QDs. For instance, the SK-Hep1 cells uptake QDs via clathrin-mediated endocytosis and caveolae-mediated endocytosis [121]. Unique cells such as alveolar macrophages use phagocytosis, energy-dependent (clathrin-mediated) endocytosis, and caveolar-mediated endocytosis to internalize graphene QDs [126]. Furthermore, the presence of serum in media was also found to have an influence on QDs’ endocytosis route. HeLa cells in media containing serum take up QDs via clathrin-mediated endocytosis, while in serum-lacking media HeLa cells seem to prefer caveolae-mediated endocytosis [119]. Carbon-based QD uptake by monocyte-derived dendritic cells revealed a size-dependent factor. Smaller-sized GQDs are more readily internalized via the dynamin-independent and cholesterol-dependent pathway, while larger GQDs are internalized in smaller quantities and only via the dynamin-independent pathway [127]. Thus, the evidence supported that the QD mode of entry can vary between cell types, QD types, and media composition. Nevertheless, clathrin-mediated endocytosis remained the main pathway for QD uptake in mammalian cells.

After entry, the subcellular distribution of QDs in cells was found to be dependent on various factors such as QD type and conjugated ligands. CdSe/ZnS QDs were mainly spotted in the lysosome, while a minute amount of QDs is found in the mitochondria, the endoplasmic reticulum, as well as the Golgi [119]. Cadmium-based QDs with negatively charged carboxylic ligands were more likely to be delivered to the endosome and the lysosome in a larger quantity, while QDs with a positively charged internalized amine ligand were found exclusively in the lysosome in lower amounts [128,129,130]. For indium-based QDs, such as InP/ZnS QDs, both carboxylic and amino ligands showed a higher internalization rate at a low treatment concentration in lung cancer cells (HCC-15) and alveolar type II epithelial cells (RLE-6TN) than QDs with a hydroxyl ligand. HCC-15 cells also had a higher QD uptake rate compared to RLE-6TN cells [131]. Similarly, when comparing HeLa cells with ML-1 cells, it was shown that HeLa cells had higher QD metabolism compared to ML-1 [132]. Thus, it hinted that QD uptake rate and post-internalized distribution are dependent on QD type, charge, and type of cells tested. Compared to Cd-based QDs, there is a lack of research on InP-based QDs’ intercellular trafficking route in mammalian cells.

A study on the trafficking route of carbon-based QDs found that CQDs are located at the cytoplasm of rat kidney cells six hours after treatment when treated with 3 mg/mL, while a higher concentration treatment of 6 mg/mL results in the accumulation of CQDs in the nucleus [133]. Likewise, in alveolar macrophages, animated graphene quantum dots (AG QDs) were also found in large amounts in the cytoplasm and the nucleus [126]. The agglomeration in the cell’s nucleus was also previously observed in carbon nanotubes [134], raising the question if this phenomenon is common among carbon-based nanoparticles. It was shown that the internalized QDs are pumped out of cells by exocytosis. The QDs undergo exocytosis using the ER/Golgi and lysosome pathway [119].

### 3.2. Impact of QDs on Cellular Level

In the past, cadmium-based QDs were thought to be highly toxic for cells. It was shown that CdTe QDs caused an increase in ROS (reactive oxygen species), leading to apoptosis of mouse liver cells, AML12 [135]. Similarly, in HeLa cells, QDs were also found to induce apoptosis and inhibit proliferation. Interestingly, the same study showed that green Cd quantum dots were more toxic than yellow QDs, indicating a size-dependent mechanism in QD toxicity [136]. CdTe QDs were also found to alter the mitochondrial membrane potential, increase the Ca^2+^ level, and cause enlargement of the mitochondria. The combination of these factors affected the integrity of the mitochondria, resulting in leaking of cytochrome c. Additionally, this study was consistent with other studies in which the treatment of cadmium ions in the form of CdCl_2_ had less overall cytotoxic impact than CdTe QDs, which supports that cadmium ion leakage is not the sole cause of cadmium-based QD toxicity [55]. Furthermore, cadmium-based QDs also showed adverse effects on neurons. It was found that CdSe QDs caused shrinkage in the cell membrane, elevated the Ca^2+^ level, and significantly reduced the cell viability of hippocampus neuronal cells [137]. Overall, it seems that cadmium-based quantum dots greatly induced toxicity against mammalian cells.

As a proposed alternative for Cd-based QDs [138,139,140], the cytotoxicity of InP on mammalian cells has been an area of interest for many researchers. When comparing the effect of CdSe QDs with InP QDs on normal human bronchial epithelial cells, it was found that CdSe QDs significantly elevated ROS generation, caused more cells to undergo necrosis, significantly reduced cellular metabolism efficiency, and severely damaged DNA. In contrast, at the same tested concentration, the ROS levels of InP-treated cells were not different from the control group. Rather, InP decreased the amount of cells undergoing necrosis, improved mitochondrial function, and did not induce significant DNA damage [141]. Consistently, several studies have suggested that treatment of InP/ZnS was not greatly toxic [142]. It was found that in five different lung-derived cell lines (PC12, KB, MDA-231-MB, NIH3T3, and B16), InP/ZnS QDs did not significantly reduce cell viability until a high concentration of 500 nM, with the exception of the B16 cell line in which the viability was significantly reduced at 100 nM concentration [131]. Thus, according to current data, it seems that InP QDs could be a safer alternative to cadmium-based QDs when assessed at the cellular level. However, most recent studies comparing Cd-based QDs and InP-based QDs in HeLa cells revealed that CdSe/ZnS QDs and InP/ZnS QDs exerted similar toxicity effects. Furthermore, the treatment of 167 µg/mL of InP/ZnS QDs in HeLa cells significantly elevated ROS levels and induced apoptosis [143], highlighting the need for further comparison between the two QD types prior to making a conclusion regarding InP-based QD safety.

Due to their recent adoption, not many studies have investigated carbon dots on mammalian systems on a cellular level, however, most existing data suggest that carbon dots are relatively non-toxic compared to their metal-containing QD counterparts [144,145]. On the other hand, it was observed that for the two cell lines NIH/3T3 and L929, carbon dots had different effects. With NIH/3T3, QDs with an ammonium ligand were found at the nucleus of the cell. However, this did not significantly reduce viability or damage DNA. In contrast, the presence of carbon dots in the nucleus led to significant cell viability reduction [146]. Thus, we can conjecture that the toxicity of carbon dots varies depending on the types of interacting cells. Therefore, more studies with different cell lines should be conducted to assess the impact of carbon dots on mammalian cells overall.

### 3.3. Effect of QDs at the Tissue and Organismal Level

While most published studies on the toxicity of QDs have focused on cellular damage, the toxicity of Cd and InP QDs on organ and tissue damage has drawn increasing attention recently. Several studies demonstrated that Cd and InP QD accumulation occurs in organs and tissues. A study conducted by Xiong et al. explored the functional and structural changes in an in vitro human airway tissue model. They found that the acute toxicity of CdSe/ZnS QDs caused aberrant mucin expression and secretion, impaired cilia functions, and caused squamous differentiation, oxidative stress production, and pro-inflammatory cytokine release, which led to impaired dysfunction of the mucosal barrier in the lung [147]. Moreover, lung functions can also be affected by InP/ZnS QDs. Compared to InP/ZnS QDs coated with a -COOH group, InP/ZnS QDs coated with an -NH_2_ group weakened the stability of the lung surfactant mixed monolayer dipalmitoyl phosphatidylcholine/dipalmitoyl phosphatidylglycerol (DPPC/DPPG). The compromised pulmonary surfactant monolayer could potentially lead to lung damage. These results indicate that the difference in surface functional groups on InP/ZnS QDs also contributes to their toxicity in the lung [148].

It has been reported that Cd QDs cause chronic toxicity in the kidney and liver. When children, adolescents, and adults were exposed to Cd QDs, the Cd QDs in urine were significantly positively associated with markers of kidney damage in all age groups, which was indicated by increased levels of N-acetyl-β-D-glucosaminidase and β2-microglobulin. In addition, the same study showed that low levels of Cd QDs in all age groups resulted in adverse tubular renal effects [149]. Another study investigated the toxicity effects of QDs with a Cd/Se/Te core and ZnS shell in mice, and the results showed that there was a reduction in renal function due to alterations in proximal convoluted tubule cell mitochondria at 16 weeks after the treatment with the QDs. Disorientation and reduction in mitochondrial number, mitochondrial swelling, later compensatory mitochondrial hypertrophy, and mitochondrial hyperplasia were observed after 16 weeks of the treatment with the QDs [150]. Additionally, degeneration of mitochondria and degeneration of cytoplasmic materials were observed within 16–24 weeks of the treatment. These results suggested that mitochondria in the kidney cells might be the target of Cd/Se/Te QDs coated with ZnS shell layers [150]. Moreover, Cd QDs can cause toxic effects on the liver. Infiltration of the liver parenchyma with lymphocytes, as well as fibrosis, microvascular steatosis of the hepatocytes, and hepatocellular micro-bubble fat degeneration were induced when the liver was exposed to Cd QDs. Importantly, Wang et al. also found that Cd QDs induced acute or chronic toxic effects in the liver, lungs, kidneys, and bones, and that then the toxicity can be absorbed into the blood from the lungs and gastrointestinal tract, which affects the blood system [151].

Some data have suggested that InP-based QDs have a lower toxicity effect compared to cadmium-based QDs at the tissue level. It was shown that in mice, inhaled InP/ZnS QDs pass through the blood barrier and accumulate in major organs, however, no major abnormalities were found in major organs. There was also no alteration in body weight and RBC count, only slight changes in WBC composition [152]. Similarly, when InP/ZnS QDs were injected in mice, it was found that QDs accumulated in major organs for an extended period of time, yet this did not result in major abnormalities [153]. In an 84-day in vivo study of BALB/c mice, it was revealed that a high dosage of InP/ZnS treatment had no toxicity effect on mice within the tested time [154]. However, even longer testing-time studies need to be conducted to ensure that InP/ZnS QDs exposure does not leave any latent detrimental consequences. As a contradiction to the low toxicity data of InP-based QDs, a study investigating the effect of InP/ZnS on mice oocytes revealed that InP/ZnS exposure causes prolonged oocyte maturation [155], indicating InP-based QDs may affect germline cells.

In vivo imaging studies of QDs in mice revealed a surface-coat-dependent distribution in major organs. Ballou et al. compared the distribution of cadmium selenide zinc sulfide quantum dots coated with four types of ligands: amphiphilic poly(acrylic acid) polymer (amp QDs); methoxyl-terminated poly(ethylene glycol) amine 750 (mPEG-750); methoxyl-terminated poly(ethylene glycol) amine 5000 (mPEG-5000); and carboxy-terminated poly)ethylene glycol) amine (COOH-PEG-3400). Post intravenous (IV) injection, the team observed QDs in the circulatory system and ligand-dependent distribution of QDs in the liver, skin, bone marrow, and lymph node; mPEG-750 QDs, amp QDs, and COOH-PEG-3400 QDs were not detected in the circulation 1 h post injection, while mPEG-5000 remained circulating for at least 3 h post injection. QDs deposited in the skin cleared after 24 h. Most QDs remained in the liver and other major organs for at least a month and could still be detected 133 days post initial IV injection [156]. Similarly, Fischer et al. also observed ligand-dependent distribution of QDs in rats. They found that 99 percent of quantum dots conjugated with bovine serum albumin (QD-BSA) were allocated to the liver and a small percentage were divided among the spleen, kidney, and bone marrow. On the other hand, only 44 percent of quantum dots conjugated with lysine-linked mercaptoundecanoic acid (QD-LM) were distributed to the liver and the rest were spotted in the lung and kidney. No QDs were found in the feces and urine samples of treated mice up to 10 days post injection, thus they concluded that QDs were sequestered and not excreted [157]. Consistently, a study found evidence of InP/ZnS QD degradation by the liver 40-days post IV injection [158]. The same study found no significant organ damage or inflammation after four weeks of QD treatment.

## 4. Quantum Dots in Fungal Cells

### 4.1. Interaction and Intracellular Trafficking

As decomposers and nutrient recyclers [159,160,161,162], fungi are an important component of our ecosystem. Therefore, before applying QDs in mass industrial-produced products, it is important to assess how QDs from discarded products will interact and affect fungal systems. For fungal cells, the rigid cell wall is a crucial factor when it comes to QD interactions. In a recent study, Qdots 625 ITKTM (QDs) were engineered on the surface of the budding yeast *Saccharomyces cerevisiae* by covalently binding yeast surface protein with the -SH group of modified QDs to investigate the interaction between QDs and the yeast surface. Results showed that Qdots 625 ITKTM (QDs) on the cell wall of the mother cell remained on its progeny for up to two generations. Although QDs remained attached to the cell surface for some time, there was no difference in yeast surface morphology when examined with an electron microscope. This study also found that engineered QD attachment on yeast surface did not affect the growth and viability of yeast [163]. In contrast, a different study found that small, free CdTe QDs, around 4.1–5.8 nm quantum dots, readily internalized into fungal cells and induced cytotoxicity by breaking cell wall components as well as causing apoptotic blebbing in the cytoplasm [52]. Consistently, recent data found that in the ascomycete fungus *Fusarium oxysporum*, CdSe/ZnS QDs easily internalized and uniformly distributed throughout hyphae after 3 h of incubation. It was also observed that QDs formed large defined clusters inside fungal cells after 16 h of incubation. Afterward, the team removed QDs from the media by washing cells and placing them in a QD-free media. Four hours after QDs were removed, a small amount of quantum dots remained inside the cells, while large aggregates were found on the cell surface and media. This led the team to hypothesize that internalized QDs are eventually secreted out, likely by exocytosis, but the exact mechanism of QD release remained unclear. Furthermore, the results in this study showed that only a high concentration, 500 nM of CdSe/ZnS QDs, compromised the growth and germination of *Fusarium oxysporum* [52]. Overall, in these findings, we could conjecture that internalized QDs had more effect on yeast viability compared to surface-attached QDs. Consistent with data seen in mammalian systems, QDs are also exported out of the cell by exocytosis, yet the exact exit pathway in yeast remains a mystery. To the best of our knowledge, the current data only indicate the uptake of QDs by fungal cells, however, the endocytic route and QDs’ exact intracellular trafficking are unknown. There is also a lack of studies demonstrating the trafficking of other types of QDs, such as InP-based and carbon-based quantum dots in yeast. This presents a knowledge gap in the field that requires immediate attention and research.

### 4.2. Toxicity Effect of QDs on Fungal Cells

Similar to the mammalian system, the protective zinc shell was also found to be effective in limiting QD toxicity in fungi. As demonstrated in a study, yeast cells treated with core-type CdSe QDs were found to have a significant reduction in mitochondrial membrane potential and acquired severe cell wall damage, while treatment of core–shell structure CdSe/ZnS QDs showed less impact on yeast cells overall [164]. The lower toxicity effect in the presence of a ZnS shell suggested that core leakage seems to be a major mechanism in QD toxicity. Interestingly, further studies show that the release of cadmium ions seems to be only one of the factors contributing to QD toxicity. In *Saccharomyces cerevisiae*, a systemic knockout mutant screening identified 114 KO mutant strains to develop tolerance to the presence of CdS QDs. These mutant strains were then tested with cadmium ions in the form of CdSO_4_. The results showed that there were only 11 CdS-QD-tolerant mutant strains showing resistance to cadmium ions, supporting that the cellular response to cadmium ions and cadmium-based quantum dots is different [165]. Thus, the toxicity of CdS QDs was not solely due to the release of cadmium ions but rather resulted from the interaction with QDs.

Another group of researchers demonstrated the impacts of CdS QDs against *Saccharomyces cerevisiae*. CdS QDs were found to induce the deletion of genes associated with stress response, metabolic processes, mitochondrial organization, DNA repair, and cellular transportation. Exposure to CdS QDs also led to an increased level of reactive oxygen species, reduced oxygen consumption, lowered both reduced and oxidized glutathione levels, and altered the mitochondrial membrane potential and mitochondrial morphology [166]. Although core–shell quantum dots are considered to cause less cellular damage, treatment of CdSe/ZnS QDs still altered the expression of many genes in *S. Cerevisiae*. Using RNA sequencing, it was found that most upregulated genes are associated with cellular component regulation, rRNA metabolic processes, macromolecule methylation, maturation of large and small subunits of ribosomes, and DNA replication in the cell cycle. Downregulated genes include genes involved in oxidation–reduction processes, small-molecule metabolic processes, proteolysis, transmembrane import and transportation, chemical responses, and electron transport chains [167].

For the mammalian cellular system, the majority of available data suggested that InP-based QDs are less toxic than Cd-based QDs. However, this was not observed in the fungal system. In 2021, Horstmann et al. were the first to compare the transcriptome of *S. cerevisiae* exposed to cadmium-based QDs (CdSe/ZnS QDs) to indium-based QDs (InP/ZnS QDs). They found that InP/ZnS inhibited yeast growth when treated at a high concentration of 100 µg/mL, while CdSe/ZnS prolonged the lag phase of yeast cells, out-compromising the final optical density at the same dosage. Furthermore, the team measured the ROS level of QD-treated cells to examine whether the inhibitory effect of InP/ZnS QDs was due to increased ROS level. They found that the ROS level in InP/ZnS-QD-treated cells decreased, while a significant elevation in ROS level was observed for CdSe/ZnS-treated cells. Furthermore, the team’s RNA-seq analysis revealed that InP/ZnS QD cells showed to have an increased expression in genes associated with antioxidant defense and peroxisome structure, while the expression of genes associated with metabolic activities was significantly decreased. On the other hand, CdSe/ZnS QDs altered genes associated with protein metabolic processes, cell wall organization, and cellular homeostasis. Interestingly, both QDs changed the level of genes associated with transmembrane transport and translation [76]. Unlike the increasing trend in the research effort on indium-based QDs in the mammalian system, available data regarding their interaction with the fungal system are limited. As the initial comparison of the two quantum dots points out the difference in the cellular response to each QD type, it is crucial to investigate the possible effect indium-based QDs have on fungal populations as we examine the possibility of using them as an alternative to cadmium-based QDs.

A few recent studies have found carbon dots to be a promising candidate for antimicrobial and antifungal drugs [168,169,170,171,172,173]. It was found that while carbon dots were non-toxic to the human cells HCT-116 and TM4 at a low concentration of 3.5 mg/mL, they had dose-dependent antimicrobial activity on fungal cells [173]. Similarly, nitrogen-doped graphene quantum dots (NDGQDs) significantly inhibited the growth of several fungal strains, including *P. citrinum, C. albicans, and Ammophilus fumigatus*, while exerted minimal impact on mouse fibroblasts cells [168]. Additionally, combining high photothermal light and carbon-based quantum dots was found to enhance QDs’ antifungal activities [169]. All this evidence suggests that fungal cells are more sensitive to carbon-based quantum dots compared to mammalian cells, and thus could potentially become effective antimicrobial agents. It is worth noting that the above studies only focused on the viability of fungal cells, yet the mechanism by which carbon dots reduce fungal viability remains to be understood. Future research regarding the specific impact of carbon-based QDs on fungal transcriptomic, proteomic, and metabolic changes is greatly needed.

While the impacts of different types of QDs are under investigation, it is important to also consider examining the interaction between QDs and fungal cells, as well as the mechanisms that induced these cellular responses. To the best of our knowledge, currently, there are very few studies focusing on the specific QD–fungal cell interactions, the QD intracellular trafficking pathway in fungal cells, QD interactions with specific organelles, and how these interactions contribute to QD toxicity in fungal cells (Table 2). This knowledge would allow us to further understand the mechanisms of QDs’ toxicity and develop strategies to limit their toxicity for safer QD applications, therefore this could be the target of future research.

## 5. Quantum Dots in Plants

### 5.1. Interaction and Intracellular Trafficking

Plants are essential to life on earth. Therefore, many studies have been conducted to understand the behavior of quantum dots in various plant species (Table 3). Like in yeast, the plant cell wall is a key component when it comes to the internalization of QDs. When studying the interaction between CdSe quantum dots and the plant cell wall isolated from conifers, it was shown that QDs bind mainly to cellulose and lignin. Furthermore, the environmental condition was also found to affect the type of interactions between QDs and the plant cell wall. In dry conditions, interactions are primarily hydrophobic interactions, while in wet conditions, some hydrophilic interactions also play a role [174]. It was found that fluid-phase endocytosis is one of the important mechanisms by which plant cells uptake CdSe/ZnS QDs, as evidenced by the overlapping of CdSe/ZnS QDs with the endocytic marker, dextran Texas Red [175]. While other endocytic routes used in QD uptake are unknown, the charge of QDs has a large impact on QDs’ internalization rate. When comparing CdS quantum dots coated with a negatively charged ligand, carboxylic acid, and a neutral charge ligand, mercaptopropionic acid (MPA), it was found that the uptake of QDs with the carboxylic acid ligand was 3.5 times slower than the neutral charge ligand. However, a large amount of both QD types were found agglomerated at the surface of the cells instead of internalized [176]. After the uptake, QDs were found to be retained in membrane-bound vesicles in cytoplasmic space for an extended amount of time [175]. It was reported that CdSe/ZnS QDs reside in both the cytoplasm and the nucleus of plant cells after 48 h of incubation. The uptake of QDs led to a dose- and time-dependent increase in plant cell ROS production [81].

It was also shown that QDs can enter whole plants and seedlings. For whole plants, QDs enter using multiple plant sites, including plant leaf and root [177,178,179,180,181]. It was suggested that after accumulating on the surface, quantum dots internalized into the plant via the apoplastic pathway through plasmodesmata, then traveled towards different organelles and accumulated in the aerial tissue [182]. For in vivo studies on the uptake of QDs in plants, the QD ligand once again showed to be important in the internalization pathway and distribution. Quantum dots with glycine and a mercaptoacetic acid ligand accumulate on the cell walls and then later enter the cells via the apoplastic pathway and reside in major organelles. QDs coated with polyvinylpyrrolidone (PVP), on the other hand, entered and accumulated in major organelles, and later translocated to the shoots via the symplastic pathway (moving from cell to cell through the cytoplasm) [183].

### 5.2. Toxicity Effect of QDs on Plants

A study showed that the uptake of quantum dots by plants majorly affects their oxidative stress process, possibly due to dose-dependent retention of CdSe QDs. The highest amount of concentration used in this study, 50 nM, was shown to cause the highest level of QD retention and ROS production [182]. Another group of researchers compared the toxic effects of CdS QDs with bulk Cd^2+^ ions on soybean plants and revealed that some alterations such as stress response in soybean were similar in both CdS QDs and cadmium ions. In addition, CdS QDs seem to also alter some metabolic processes negatively, including the citric acid cycle and glycolysis [183]. This resembles the finding in the fungal system that the toxicity of QDs is not solely due to toxic ion leaching, but other components of QDs as well. Moreover, different types of QDs were shown to have different effects on plants.

Since GQDs with a negatively charged hydroxyl ligand readily enter plant cells, they cause higher levels of photosynthesis inhibition, and were found to activate peroxidase, super dioxide dismutase, and catalase compared to GQDs with a positively charged amine ligand [184,185]. Similarly, the toxicity effect of GQDs on lettuce has been shown to be dependent on the functional group of the QDs’ ligand. In 50 mg/L of GQD treatment, the three examined amine, carboxyl, and hydroxyl functional groups caused similar levels of reduction in lettuce root length (around 36–41%). In a higher concentration of 100 mg/L, hydroxylated GQDs reduced lettuce root length by 85%, while GQDs with a carboxylic and amide group resulted in the reduction rate remaining around 32–41%. The shoot length was found to only be shortened by GQDs with a hydroxyl functionalized group. Additionally, the ion leakage level of the cell membrane in root cells was also higher with hydroxyl GQDs compared to amine and carboxylic GQDs [186]. This evidence collectively showed that graphene quantum dots with hydroxyl surface functional groups have a higher toxicity effect on the plant system compared to amine or carboxylate surface ligands. Carbon dots are also shown to have mild cytotoxicity effects in plants. In Arabidopsis seedlings, CDs caused inhibition of root growth by altering the expression of genes related to DNA damage repair and the cell cycle [187]. High concentrations of CDs decreased root and shoot growth, as well as increased activities in enzymes such as catalase and superoxide dismutase [180].

## 6. Interactions of QDs with Different Cellular Components

### 6.1. Interaction with Lipid Membrane

To understand QDs’ impact on lipid membranes, Wlodek et al. studied the interaction between 4.9 nm CdS QDs and supported lipid bilayers (SLB) [188]. The in vitro cell-free reconstitution experiment data revealed that after 3 h of incubation, SLB with QDs showed a reduction in thickness, which they attributed to the titling of a hydrophobic tail to interact with QDs as QDs begin to protrude themselves into the phospholipid bilayers. The thickness of SLB increases over the incubation time as the lipid bilayer rearranges to accommodate QDs in the hydrophobic region of the phospholipid bilayer. The inner bilayer leaflet of SLB steadily regains thickness over time, as it expels QDs to the outer bilayer leaflet, which is more likely to be deformed due to less enhanced lipid packing. The insertion of QDs into the lipid bilayer seems to be dependent on the QDs’ size. When attempted to incorporate CdSe QDs into the lipid bilayer of a unilamellar vesicle containing dioleoyl phosphatidylcholine (DOPC) lipid molecules, QDs with a diameter of 1.05 nm–1.65 nm were readily inserted between the lipid bilayer in the hydrophobic tail region, while QDs with a diameter around 2.5 nm were only found aggregating outside of the vesicle. However, in a vesicle containing a DOPG lipid molecule, which is characterized by having a negatively charged and larger head group area, QDs with a diameter of 2.5 nm were able to be incorporated into the lipid bilayer of the vesicle [188]. As components of the lipid bilayer rearrange to make room for QD insertion, whether the integrity of the lipid membrane can be retained is in question. One group of researchers investigated the stability of the membrane when interacting with QDs. SLB with DOPC lipid molecules were exposed to CdSe/ZnS QDs (core size 3.3 nm) with a positive poly-diallyl dimethylammonium chloride (PDDA) coating. PDDA-coated QDs readily interacted with the lipid membrane and caused rearrangement of the lipid bilayer’s structure. PDDA QDs also led to the disappearance of the liquid-ordered domain after 15 min of interaction, which caused destabilization of the lipid membrane [189]. This result suggests that in interacting with the lipid bilayer, certain quantum dots could insert themselves in between the lipid bilayers and become transported into the vesicle. However, this may compromise the stability of the membrane layers, as it temporarily compromised their structure due to the insertion of QDs. As a confirmation of this phenomenon, a study on the uptake of CdSe/ZnS QDs with a zwitterionic thiol ligand D-penicillamine (DPA) coat by RBCs, which is incapable of performing endocytosis, revealed that QDs could indeed be internalized into cells by protruding the lipid bilayer of the membrane. At one hour of incubation, QDs were found at the membrane layer of RBCs, likely between the hydrophobic tail regions of the lipid bilayers. Later, small clusters of QDs were spotted inside RBCs, and these clusters increased in size and fluorescence over time. The passage of QDs into RBCs did not induce hole formations in the lipid membrane, rather, the lipid conformation was compromised and found to be more flexible to allow QDs to pass through [190].

In vitro studies of the interaction of QDs with the lipid membrane are mostly focused on cadmium-based QDs. The lack of data regarding the lipid membrane’s interaction with other types of QDs highlights the need for further studies.

### 6.2. Interaction upon Internalization

Studies have found that after uptake by receptor-mediated endocytosis, most QDs are naturally retained inside the endosome for a lengthened period. Rather than being released into the cytosol, internalized QDs are trapped in a membrane-bound vesicle of the endosomal and lysosomal trafficking pathway [191]. Since QDs mainly use endocytosis as a main mode of entry and are encapsulated in a membrane-bound vesicle, QDs often are not found in the cytosol. The inability of QDs to reach the cytosol limits their application as a cytosolic probe, thus new strategies have been developed to effectively deliver QDs into cytoplasmic space. Delehanty et al. found that using established methods such as using sucrose and chloroquine to promote the release of QDs results in aggregation of QDs in the cytosol and leads to severe cellular damage. A better strategy is to incorporate an amphiphilic polymer agent which promotes rapid QD uptake, followed by a slower endosomal escape into the cytosol peaking around 48 h after the initial uptake, which exerts less cellular damage compared to other methods [192]. Some studies suggest using a cationic shell to induce low pH and disrupt the endosomal membrane, helping QDs to translocate to the cytosol [193]. A better strategy is to incorporate positively charged arginine-terminated gold, which passively translocates through the cell membrane by facilitating a temporary opening in the membrane to carry QDs across the plasma membrane directly to the cytosol.

It is well known that post-endocytosed contents targeted for degradation, including nanoparticles, are carried by the late endosome which then directly fuses with the lysosome [194]. The targeting of QDs to the lysosome is consistent with the trafficking of other nanoparticles, where most nanoparticle content aggregates within the lysosome which plays an important role in NP cytotoxicity [195,196,197]. Similarly, QDs’ interaction with the lysosome reportedly triggered ROS and ROS-induced autophagy, which contributed to the toxicity of QDs on mammalian cells. Evidently, when the lysosomal activity is inhibited, cells experience less cellular damage resulting from CdTe/CdS 569 QD treatment compared to cells with normal lysosomal activity [198]. This further emphasizes the importance of the lysosome in both NPs and QDs toxicity.

Recently, the question of whether QDs are targeted to the nucleus has been raised by many researchers. This information is important to both understanding the toxicity mechanism of QDs and their potential as a nuclear-targeting drug delivery system. Maity et al. used CdSe/ZnS QDs with a carboxylic ligand coated with a carbodiimide linker and nuclear localization signal (NLS) peptide to investigate their effect on the uptake and intracellular localization of QDs in HeLa cells. They found that QDs with densely attached NLS promoted the endocytosis of QDs and caused accumulation of QDs near or inside the nucleus. Interestingly, all tested QD groups (bare CdSe/ZnS QDs, CdSe/ZnS with carbodiimide linker, CdSe/ZnS QDs with linker and NLS) were able to escape the endo-lysosome system, as demonstrated by the lack of colocalization with late endosome and the lysosome tracker dye 12 h post QD treatment [199]. Another study using alveolar macrophages and graphene QDs revealed that graphene QDs were found in the endo-lysosome, the mitochondria, and the endoplasmic reticulum after QDs were endocytosed. However, after 24 h of incubation, most QDs were found accumulating at or near the nucleus. When exposed to 50 μg/mL of GQDs for 24 h, the inner nuclear envelope shrinkage was observed, while a higher concentration of GQDs (100 and 200 μg/mL) caused the irregular appearance of the nucleus. It was also observed that AG QD exposure led to DNA cleavage, likely caused by oxidative stress, direct QD–DNA interaction, and apoptosis [125].

### 6.3. Interaction with Proteins

Due to the endless potential of quantum dots as biological probes, understanding quantum dots’ interaction with various proteins is essential. The interaction between quantum dots and proteins seems to be dependent on each protein’s structure as well as the concentration of protein exposed to QDs. For lysozyme, CdTe QDs induce changes to the 3D protein structure, which leads to degradation of the alpha helices’ hydrogen bonding. The measurement from FTIR (Fourier-transform infrared spectroscopy) of the CdTe QDs’ protein complex showed a prominent emission peak in the proline region as protein concentration increased, suggesting that CdTe QDs mostly interact with proline on the alpha helices of lysozyme. In hemoglobin, CdTe interacts with regions containing amide, with more hydrogen binding formed between the porphyrin ring of the heme region with CdTe QDs. For a lower BSA protein concentration (10–100 µg/mL), CdTe QDs showed to mainly interact with serine and tyrosine regions. However, the interaction between tyrosine and CdTe QDs seems to be impeded with higher protein concentration (200 µg/mL). When comparing these proteins, hemoglobin was the most stable protein out of the three tested post QD exposure, due to the porphyrin ring structure, while BSA was the most unstable because it underwent conformational changes starting at 25–50 µg/mL of protein [200].

In a quest to look for the specific protein property that QDs interact with, Lu et al. looked at the interaction between CdTe QDs capped with mercaptosuccinic acid (MSA), and the three proteins glucose oxidase (GOD), hemoglobin (Hb), and cytochrome c (Cyt C). Two forms of QDs were tested, immobilized QDs attached to a gold plate and aqueous QDs. They found that for immobilized QDs, cytochrome c seems to have the greatest binding affinity to QDs, followed by hemoglobin, and lastly glucose oxidase. However, for aqueous QDs, more Hb is bound to the surface of QDs. Furthermore, aqueous QDs caused aggregation of Hb proteins, forming micro QD-Hb clusters, which was not seen in the other two proteins.

The pH seemed to play a key role in QD–protein complex affinity, with low pH promoting higher binding affinity of all three proteins, suggesting that QDs interact with proteins by electrostatic interaction. The weak binding affinity of the glucose oxidase protein, which has a predominantly negative charge, compared to cytochrome c, which is a predominantly positively charged protein, also hinted at the importance of protein surface charges when interacting with QDs. Similarly, in the case of amine–QD and carboxylate–QD interactions with plasma protein, it was found that the protein–QD complex had a higher affinity at lower pH, around 5.5. The adsorption of plasma protein by the two QDs formed a coronated protein film that was mostly composed of albumin. Amine QDs reportedly accumulated significantly more corona protein compared to carboxylate protein [201]. From the interaction between quantum dots and the stated above protein, it seems that the QD–protein interaction could also play a role in QD toxicity. The formation of QD clusters could sequester proteins, leading to a reduction in optimal protein levels for essential biological processes. Particularly, the deficiency of cytochrome c causes mitochondrial dysfunction, negatively impacts cellular metabolic pathways, and alters other essential biological processes [202,203,204,205,206]. Thus far, only a few types of plasma proteins have been tested. More plasma proteins could be assessed to determine altered protein conformation and function as a result of exposure to QDs. Moreover, as proteins are an essential part of cells and tissues, studies of the interaction between cellular proteins and QDs are highly needed to understand the overall impact of QDs.

## 7. QD-Induced Proteomic Changes

Recently, more attention has been given to the toxic effects of QDs on protein expression. A study conducted by Xu et al. revealed that 50 nM CdTe QDs coated with mercaptoacetic acid (MPA) inhibited the protein expression of glutathione S-transferase (GST) in *E. coli* [207]. This result suggested that the high concentration of MPA-CdTe QDs caused serious toxicity in *E. coli*. They show that the inhibition of the protein expression of GST was because MPA-CdTe QDs affect the proliferation of *E. coli*. Another study investigated the proteins in yeast in response to the treatments with CdS QDs, and the results showed that proteins involved in oxidative stress were significantly affected. Moreover, key proteins that were involved in the main pathways were grouped. Among these pathways, three enzymes involved in the glycolysis pathway were upregulated at 9 h and most enzymes associated with the glycolytic pathway were downregulated at 24 h, including acetyl-coenzyme A synthetase 1 (Acs1), dihydrolipoyl dehydrogenase (Ldp1), pyruvate decarboxylase isozyme 5 (Pdc5), mitochondrial potassium-activated aldehyde dehydrogenase (Ald4), and NADP-dependent alcohol dehydrogenase 2 (Adh2). Additionally, key proteins involved in the oxidative phosphorylation chain, the ATP-dependent molecular chaperone Hsc82, and protein folding and ubiquitination in the endoplasmic reticulum were also downregulated after 24 h after CdS QDs treatment. Compared to the results of 9 h, oxidative stress decreased, and the lethality of cells increased at 24 h. There was a metabolic shift from respiration to fermentation when cells were treated with QDs [75]. For plants, stress-related and hormone-regulated proteins were found to be downregulated in response to CdS QDs, which were unexpected results of QD treatment [208].

Proteomics involves the identification and quantification of overall proteins expressed from a cell, tissue, or organelle, thereof resulting in an information-rich landscape of modulations of proteins under specific conditions. Proteomics has been seen as an important aspect of current postgenomic biology approaches to understanding molecular mechanisms underlying normal and disease phenotypes. Particularly, it has been mainly used in the areas of cancer research, drug and drug target discovery, and biomarker research. However, there is a lack of proteomic study in the field of nanomaterials, especially quantum dots, due to the complexity of protein molecules themselves, which makes comprehensive studies of proteomes challenging. The basic interactions of nanoparticles with proteins play a central role in nanomedicine and in concerns about nano safety. Thus, we believe that a clear view of the interactions of QDs and proteins is necessary. Additionally, the study of proteomic changes by QDs together with transcriptomic analyses is limited, and the statistical correlation between protein abundance and transcription is poor, especially in eukaryotic cells. Comprehensive investigations of proteomic and transcriptomic changes by QDs could lead to a better understanding of molecular, genetic, and physiological mechanisms.

## 8. Future Research Trends

In this mini-review, we discussed several types of quantum dots as well as their interaction and toxicity towards three different living systems: mammalian, fungal, and plant. The effect of quantum dots not only varies between the living systems and cell types, but also varies depending on the treatment condition, type of quantum dots used, and the properties of each type of QDs. The complexity of the cytotoxicity mechanism of QDs showed that much more research effort needs to be made to ensure the safety of utilizing QDs in biomedical science and commercialized products. Based on the data presented in this review, we composed a list of possible research trends:I.**Tracking quantum dot subcellular trafficking and interaction with fungal cells**. Although fungal organisms are a major part of our ecosystem, the available data regarding the interaction between different types of QDs and fungal cells are minimal. In particular, QDs’ mode of entry is unclear in fungal cells. Current data show the uptake of QDs by various fungal cells; however, it is still unknown if QDs are internalized by energy-dependent endocytosis or spontaneous uptake. Furthermore, if QDs are indeed internalized by endocytosis, the main endocytic route used also needs to be identified. In addition, subcellular trafficking and post-internalization interactions in yeast are unknown. It is highly likely that the research interest will gravitate towards the distribution of quantum dots in yeast, specifically whether QDs are transported to the endosome, the ER, the Golgi, the lysosome, and the nucleus. In addition, compared to the well-studied exocytosis event in mammalian cells, our understanding of how fungal cells discard internalized QDs is also lacking. All the mentioned points are major current knowledge gaps in the field, and therefore, it could be predicted that these areas will be the trend for future research.II.**Quantum dot–protein interaction.** Due to the advancement in proteome analysis technology, proteomics has successfully captured the interest of researchers worldwide. When it comes to quantum dots, two proteomic approaches could be adopted. First, the QD-induced alteration of proteome profiles should be investigated. As we gathered data for this review, we noticed that while an abundance of transcriptomic data is available, very few studies have used the proteomic approach to study QD cytotoxicity in all three examined living systems. This type of study would also reveal post-translation alteration as a result of exposure to QDs, thus allowing a more well-rounded understanding of the impact of QDs on different living systems. Secondly, since protein is a major cellular component, the identification and quantification of QD-interacting proteins would be vastly valuable. It has been shown that QDs have relatively strong interactions with all three examined living systems. Thus, it would be interesting to assess which protein can interact with quantum dots, the mechanism of the interaction, and lastly how the interaction affects protein conformation and function. Understanding the impact of QDs on protein would provide a novel insight into other QD toxicity mechanisms, thus, it is highly likely that this will be a future trend for research.III**Epigenetics approach**. To the best of our knowledge, there is little to no data available regarding the impact of QDs on the epigenetics of cells. However, an abundance of data has suggested that epigenetic alteration is a major factor in nanoparticle toxicity [209,210,211,212,213]. Therefore, it would be novel to assess QD-induced DNA methylation and histone modification, then determine if these modifications are temporary or permanent. An assessment of the transgenerational inheritance of QD-induced epigenetic modifications would also be very interesting. In recent years, the epigenetic approach has been increasingly adopted in various toxicity studies. Thus, it would be beneficial to study the possibility that epigenetic modifications could be one of the factors that contribute to QDs’ toxicity.IV**Single-cell sequencing.** As a potential candidate for in vivo applications, it is essential to investigate how quantum dots impact different types of cells, since the characteristic, function, and response of each cell is different. Hence, single-cell sequencing could be applied to compare the differences in the genetic profile of different cell types post-QD exposure. Cells from different major organs could be harvested and evaluated for their toxicity impact, thus giving a more comprehensive view of the effect of QDs on an organismal level. This is expected to be the next step in in vivo studies on QDs.V**More advanced research in improving quantum dots to further reduce their toxicity for safe in vivo applications**. It was suggested that the addition of the protective shell and ligands on QDs’ surface reduces core material leakage, resulting in reduction in toxicity, yet some amount of leakage still occurs [23]. Thus, the development of advanced protective surface components to prevent all core leakage is essential in establishing safe QDs for large-scale adoption. Furthermore, as a potential non-toxic alternative to their metal-based QD counterpart, the evaluation of the toxicity of carbon-based quantum dots in different living systems is also highly important, particularly the effect of carbon-based QDs on the mammalian cellular system. Since some recent studies have indicated that a low level of toxicity still occurs in the presence of these carbon-based quantum dots, a further research aim would likely be to further minimize or possibly eliminate the toxicity of carbon-based quantum dots to make them more suitable for in vivo applications such as disease detection and drug delivery.

Each of the mentioned future research trends are equally essential (Figure 2). Collectively, the generated knowledge will fill in major knowledge gaps in the relevant fields and provide a well-rounded understanding of the impact of QDs on different living systems. From there, better strategies could be developed to improve the safety and efficiency of quantum dots for many applications.

## Figures and Tables

**Figure 1 ijms-23-10763-f001:**
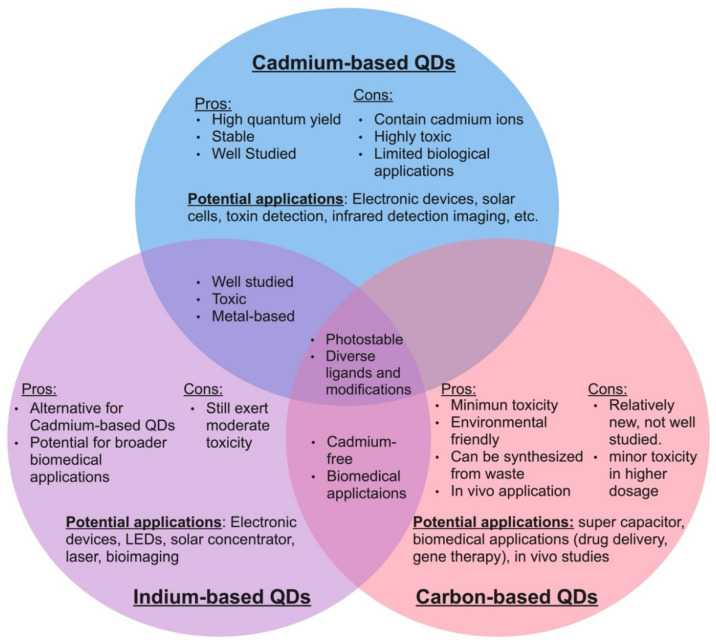
Characteristics of the three major quantum dots core types.

**Figure 2 ijms-23-10763-f002:**
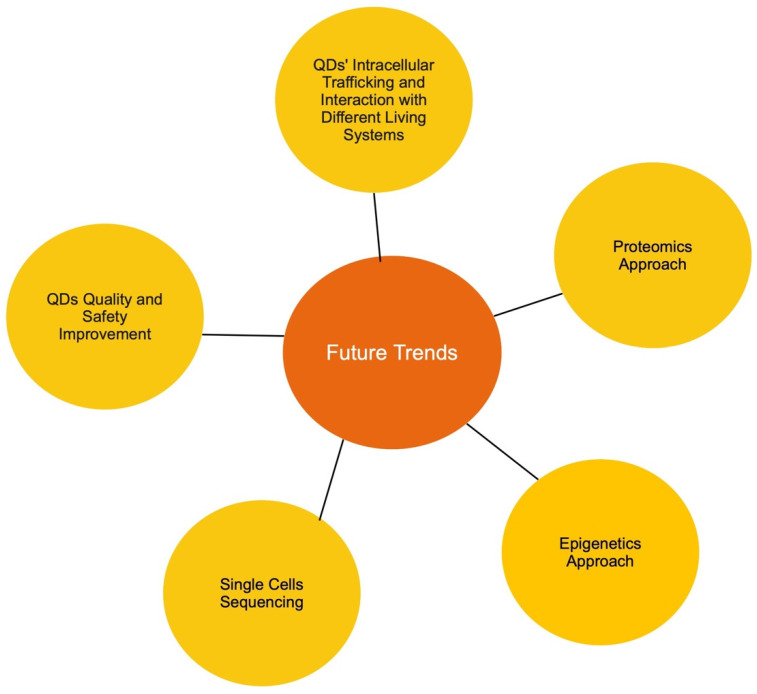
Future research trends for QD-related topics.

**Table 1 ijms-23-10763-t001:** Relevant articles cited in the mammalian system section.

Article Title	Author	Reference
Fate of CdSe/ZnS Quantum Dots in Cells: Endocytosis, Translocation and Exocytosis	Liu et al.	[119]
Clathrin-Mediated Endocytosis of Quantum Dot-Peptide Conjugates in Living Cells	Anas et al.	[120]
Rapid Intestinal Uptake and Targeted Delivery to the Liver Endothelium Using Orally Administered Silver Sulfide Quantum Dots	Hunt et al.	[121]
Quantum Dot Bioconjugates for Ultrasensitive Nonisotopic Detection	Chan et al.	[122]
Atomic Force Microscopy-Based Cell Nanostructure for Ligand-Conjugated Quantum Dot Endocytosis	Pan et al.	[123]
Cellular Uptake Mechanisms and Toxicity of Quantum Dots in Dendritic Cells	Zhang et al.	[124]
Nano-Engineered Skin Mesenchymal Stem Cells: Potential Vehicles for Tumour-Targeted Quantum-Dot Delivery	Saulite et al.	[125]
Graphene Quantum Dots in Alveolar Macrophage: Uptake-Exocytosis, Accumulation in Nuclei, Nuclear Responses and DNA Cleavage	Xu et al.	[126]
Graphene Quantum Dots Suppress Proinflammatory T Cell Responses via Autophagy-Dependent Induction of Tolerogenic Dendritic Cells	Tomic et al.	[127]
Cell-Penetrating Quantum Dots Based on Multivalent and Endosome-Disrupting Surface Coatings	Duan et al.	[128]
Revisiting the Principles of Preparing Aqueous Quantum Dots for Biological Applications: The Effects of Surface Ligands on the Physicochemical Properties of Quantum Dots	Zhang et al.	[129]
Surface-Ligand-Dependent Cellular Interaction, Subcellular Localization, and Cytotoxicity of Polymer-Coated Quantum Dots	Tan et al.	[130]
Cytotoxicity of InP/ZnS Quantum Dots with Different Surface Functional Groups toward Two Lung-Derived Cell Lines	Chen et al.	[131]
Intracellular Trafficking and Distribution of Cd and InP Quantum Dots in HeLa and ML-1 Thyroid Cancer Cells	Zhang et al.	[132]
Carbon Quantum Dots from Roasted Atlantic Salmon (Salmo Salar L.): Formation, Biodistribution and Cytotoxicity	Song et al.	[133]
Uptake of Single-Walled Carbon Nanotubes Conjugated with DNA by Microvascular Endothelial Cells	Harvey et al.	[134]
Liver Toxicity of Cadmium Telluride Quantum Dots (CdTe QDs) Due to Oxidative Stress in Vitro and in Vivo.	Zhang et al.	[135]
The Future of Anticancer Drugs: A Cytotoxicity Assessment Study of CdSe/ZnS Quantum Dots	Hens et al.	[136]
Unmodified CdSe Quantum Dots Induce Elevation of Cytoplasmic Calcium Levels and Impairment of Functional Properties of Sodium Channels in Rat Primary Cultured Hippocampal Neurons	Tang et al.	[137]
An in Vitro Investigation of Cytotoxic Effects of InP/ZnS Quantum Dots with Different Surface Chemistries	Ayupova et al.	[138]
Indium Phosphide: Cadmium Free Quantum Dots for Cancer Imaging and Therapy	Chibli et al.	[139]
Interaction between Human Serum Albumin and Toxic Free InP/ZnS QDs Using Multi-Spectroscopic Study: An Excellent Alternate to Heavy Metal Based QDs	Sannaikar et al.	[140]
Quantifying Engineered Nanomaterial Toxicity: Comparison of Common Cytotoxicity and Gene Expression Measurements	Atha et al.	[141]
The Cytotoxicity Studies of Water-Soluble InP/ZnSe Quantum Dots	Kiplagat et al.	[142]
An Assessment of InP/ZnS as Potential Anti-cancer Therapy: Quantum Dot Treatment Induces Stress on HeLa Cells	Davenport et al.	[143]
Nitrogen Doped Carbon Quantum Dots Demonstrate No Toxicity under: In Vitro Conditions in a Cervical Cell Line and in Vivo in Swiss Albino Mice	Signh et al.	[144]
Normal Breast Epithelial MCF-10A Cells to Evaluate the Safety of Carbon Dots	Vale et al.	[145]
Self-targeting of Carbon Dots into the Cell Nucleus: Diverse Mechanisms of Toxicity in NIH/3T3 and L929 Cells	Havrdova et al.	[146]
The Glutathione Synthesis Gene Gclm Modulates Amphiphilic Polymer-Coated CdSe/ZnS Quantum Dot-Induced Lung Inflammation in Mice	McConnachie et al.	[147]
Effects of Carboxyl or Amino Group Modified InP/ZnS Nanoparticles Toward Simulated Lung Surfactant Membrane	Wang et al.	[148]
Kidney Toxicity and Response of Selenium Containing Protein-Glutathione Peroxidase (Gpx3) to CdTe QDs on Different Levels	Zhao et al.	[149]
Electronic Microscopy Evidence for Mitochondria as Targets for Cd/Se/Te-Based Quantum Dot 705 Toxicity in Vivo	Lin et al.	[150]
Degradation of Aqueous Synthesized CdTe/ZnS Quantum Dots in Mice: Differential Blood Kinetics and Biodistribution of Cadmium and Tellurium	Liu et al.	[151]
Biodistribution and Acute Toxicity of Cadmium-Free Quantum Dots with Different Surface Functional Groups in Mice Following Intratracheal Inhalation	Lin et al.	[152]
In Vivo Comparison of the Biodistribution and Toxicity of InP/ZnS Quantum Dots with Different Surface Modifications	Li et al.	[153]
In Vivo Toxicity Assessment of Non-Cadmium Quantum Dots in BALB/c Mice	Lin et al.	[154]
Evaluation for Adverse Effects of InP/ZnS Quantum Dots on the in Vitro Cultured Oocytes of Mice	Ye et al.	[155]

**Table 2 ijms-23-10763-t002:** Relevant articles cited in the fungal system section.

Article Title	Author	Reference
Fungal Importance Extends beyond Litter Decomposition in Experimental Early-Successional Streams	Frossard et al.	[159]
Socialism in Soil? The Importance of Mycorrhizal Fungal Networks for Facilitation in Natural Ecosystems	Van der Heijden et al.	[160]
The Missing Metric: An Evaluation of Fungal Importance in Wetland Assessments	Onufrak et al.	[161]
Hildebrand, F. Metagenomic Assessment of the Global Diversity and Distribution of Bacteria and Fungi.	Bahram et al.	[162]
Determining the Fate of Fluorescent Quantum Dots on Surface of Engineered Budding S. Cerevisiae Cell Molecular Landscape	Chouhan et al.	[163]
The Interactions between CdSe Quantum Dots and Yeast Saccharomyces Cerevisiae: Adhesion of Quantum Dots to the Cell Surface and the Protection Effect of ZnS Shell	Mei et al.	[164]
Yeast Populations Evolve to Resist CdSe Quantum Dot Toxicity	Strtak et al.	[165]
Nucleo-Mitochondrial Interaction of Yeast in Response to Cadmium Sulfide Quantum Dot Exposure	Pasquali et al.	[166]
Transcriptome Profile Alteration with Cadmium Selenide/Zinc Sulfide Quantum Dots in Saccharomyces Cerevisiae	Horstmann et al.	[167]
Preparation and Characterization of B, S, and N-Doped Glucose Carbon Dots: Antibacterial, Antifungal, and Antioxidant Activity	Ezati et al.	[168]
Green Synthesis of Multifunctional Carbon Dots for Anti-Cancer and Anti-Fungal Applications	Zhao et al.	[169]
Amine-Coated Carbon Dots (NH2-FCDs) as Novel Antimicrobial Agent for Gram-Negative Bacteria	Devkota et al.	[170]
Carbon Dots as an Emergent Class of Antimicrobial Agents	Ghirardello et al.	[171]
Antimicrobial Activity and Characterization of Pomegranate Peel-Based Carbon Dots	Qureshi et al.	[172]
One-Pot Microbial Approach to Synthesize Carbon Dots from Baker’s Yeast-Derived Compounds for the Preparation of Antimicrobial Membrane	Ghorbani et al.	[173]
Toxicity of CdTe Quantum Dots on Yeast Saccharomyces Cerevisiae	Han et al.	[52]
Meta-Analysis of Cellular Toxicity for Cadmium-Containing Quantum Dots	Oh et al.	[83]

**Table 3 ijms-23-10763-t003:** The relevant article cited in the plant system section.

Article Title	Author	Reference
Interaction of the CdSe Quantum Dots with Plant Cell Walls	Djikanović et al.	[174]
Fluid Phase Endocytic Uptake of Artificial Nano-Spheres and Fluorescent Quantum Dots by Sycamore Cultured Cells	Etxeberria et al.	[175]
Cell Wall: An Important Medium Regulating the Aggregation of Quantum Dots in Maize (Zea Mays L.) Seedlings	Sun et al.	[176]
Nanoparticle Charge and Size Control Foliar Delivery Efficiency to Plant Cells and Organelles	Hu et al.	[177]
Lipid Exchange Envelope Penetration (LEEP) of Nanoparticles for Plant Engineering: A Universal Localization Mechanism	Wong et al.	[178]
In Vivo Plant Flow Cytometry: A First Proof-of-Concept	Nedosekin et al.	[179]
The Effect and Fate of Water-Soluble Carbon Nanodots in Maize (Zea Mays L.)	Chen et al.	[180]
Uptake and Accumulation of CuO Nanoparticles and CdS/ZnS Quantum Dot Nanoparticles by Schoenoplectus Tabernaemontani in Hydroponic Mesocosms	Zhang et al.	[181]
High Efficiency Transport of Quantum Dots into Plant Roots with the Aid of Silwet L-77	Hu et al.	[182]
Surface Coating Determines the Response of Soybean Plants to Cadmium Sulfide Quantum Dots	Majumdar et al.	[183]
Effect of Graphene Quantum Dot Size on Plant Growth	Xu et al.	[184]
Surface Charge Affects Foliar Uptake, Transport and Physiological Effects of Functionalized Graphene Quantum Dots in Plants	Sun et al.	[185]
Size Effect of Graphene Quantum Dots on Photoluminescence	Liu et al.	[186]
Carbon Dots Inhibit Root Growth by Disrupting Auxin Biosynthesis and Transport in Arabidopsis	Yan et al.	[187]

## Data Availability

Not applicable.

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
