# Peer review of "Quantum Dots and Their Interaction with Biological Systems"

_ijms, 2022, doi:10.3390/ijms231810763_

Round 1

Reviewer 1 Report

This mini review summarizes recent advances in studying interactions and effects of quantum dots on living systems.  Such a review could be useful for other researchers working in the QD fields and/or looking into applying available QDs.  This justifies my recommendation to publish this manuscript but only after addressing the follow deficiencies:

(1)    The topic of QD toxicity is certainly not new and this must be acknowledged at the very beginning of the review.  Then the authors should also mention the preceding detailed and well-cited reviews on this topic such as, for example, Environ Health Perspect. 2006 Feb; 114(2) 165–172; Small, 2010 Jan;6(1):138-44, doi: 10.1002/smll.200900626, In vivo quantum-dot toxicity assessment; Chem. Commun., 2011,47, 7039-7050; Accounts Of Chemical Research, 662–671 (2013),  Vol. 46, No. 3, among some other.

(2)    While the authors provide a “laundry list” of recent observations, they miss on a clear discussion of physical factors (like size, shape, and surface charge) as well as key chemical properties that govern interactions of QDs with biological systems.  This decreases the value of the review significantly.  Also, the properties QDs are known be affected by illumination and this has to be discussed.  Also, there is no discussion of the effects of QD shape.

(3)    Regarding the section “3.3. Impact of QDs on a Tissue Level”:

I find the title to be rather confusing.  Do the authors mean “Effects of QDs at the Tissue Level”?   Otherwise, the title assumes that the is an impact of QDs on some kind of a level but which one?  Instead, the authors should discuss recent advances in understanding of QD biodistribution in animals and clearance mechanisms as those are directly related to toxicity.

(4)    (a) Finally, the authors are asked to edit the manuscript for English and style starting with the title “Quantum Dots and Its Interaction with Different Living Systems”

-          Perhaps “Quantum Dots and Their Interactions”?

-          No need for “Different”.

-          I suggest a simpler title of “Interactions of Quantum Dots with Biological Systems”

(b) I suggest avoiding “are currently highly sought after for …” and “Amid quantum dots fever” in the abstract.  The abstract should be re-written to reflect the content of the review rather than to be composed from a set of general non-descriptive statements.

(c ) What do you mean by “systems” in “mammalian system, fungal system, and plant system”?  Please be specific/descriptive.

(d) Line 227: “groups significantly affect the physiological toxicity of InP/ZnS QDs to lung

health” – please rewrite.

(e) Line 334: “ROS (Reactive Oxygen Species)”: ROS were discussed earlier in the review and this abbreviation should be spelled out earlier and listed in the Table of abbreviations.

(f) This list could be readily expanded.

Author Response

See the attached document.

Reviewer 2 Report

The manuscript is well written in all parts and contains complete information and figures. Authors should provide a complete table, including the reviewed studies, in the manuscript.

Author Response

Check the attachment.

Round 2

Reviewer 1 Report

I am satisfied with the changes made by the authors.